# Proteogenomic Approaches to Understand Gene Mutations and Protein Structural Alterations in Colon Cancer

Soumyadev Sarkar [ID]

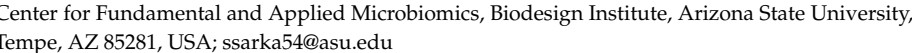

Center for Fundamental and Applied Microbiomics, Biodesign Institute, Arizona State University, Tempe, AZ 85281, USA; ssarka54@asu.edu

**Abstract:** With colon cancer being one of the deadliest and most common cancers, understanding the mechanisms behind colon cancer is crucial in improving therapies. One of the newest approaches in cancer research is the concept of proteogenomics. While genomic data is not sufficient to understand cancer, the integration of multi-omics data including proteomics in conjugation with protein modeling has a better potential to elucidate protein structural alterations and characterize tumors. This enables a more efficient diagnosis of cancer and improves remedial strategies. In this review, we aim to discuss the linkage between gene mutations and protein structural alterations that lead to colon cancer. Topics include alterations in the glycoproteome and structures of proteases that impact colon cancer development. Additionally, we highlight the importance of precision oncology with an emphasis on proteogenomic approaches, protein modeling, and the potential impact on colon cancer therapy.

**Keywords:** colon; colorectal; cancer; proteogenomics; protein; structure; modeling; function; precision; oncology

## 1. Introduction

The projected annual incidence of cancer is around 18 million, with colorectal cancers (CRC) ranking third among male cancers and second among female cancers [1]. The second most deadly cancer in the world, CRC is also the third most common cancer overall and has a high mortality rate of 9.2% of all cancer deaths [2]. More recently, the age group with higher incidence rates has fallen below the usual recommendation for starting colon cancer screenings [3]. This pattern is concerning since more colon cancers are now likely to occur in individuals that have not yet been screened and remain undiscovered for longer. In addition to being deadly and frequently occurring, CRC is a very complicated illness with many interrelated variables. The many environmental and genetic factors that affect the etiology of CRC contribute to its complexity.

Nearly 42% of all cancer cases have risk factors that may be preventable, such as smoking, eating poorly, getting too little fiber, being overweight, and not getting enough exercise [4]. However, only around 10% of CRC cases are due to family history or genetic ties, and the majority of CRC incidences are indirectly caused by a number of reasons which are primarily sporadic in combination with some hereditary factors [5]. The effect of the gut microbiota, including its activities, byproducts, and interactions with the host, on the risk of developing CRC is also intertwined with these factors. When there is dysbiosis or a lack of certain functions required for homeostasis, the microbiota can negatively or positively affect the immune system and colon [6]. Therefore, the complete picture of CRC development is probably not best captured by genomic-only techniques because CRC risk factors are complicated. Multi-omics methods are a fantastic tool for capturing many biological vantage points and can reveal new facets of complex diseases like CRC, which involve a wide variety of risk factors [7]. To progress diagnoses and the finding of biomarkers, other methods must be taken into consideration because genetics and familial history only have a minor impact on the total caseload.

Proteomics deal with the identification, localization, regulation, and quantification of proteins in a biological system. This method can be used to investigate significant protein modifications that genomic-only approaches overlook, like posttranslational alterations that significantly change a protein's functional capabilities [8]. Proteogenomics, which combines genomic techniques with a proteomics approach, has the potential to identify more protein markers for CRC, allowing for earlier identification of risk factors and the implementation of therapy or preventative measures. In order to understand the functions associated with these proteins, proteogenomics has been employed in the field of colon cancer to identify the protein abundance profiles of CRC samples and link the abundance to functional genomic data [9,10]. Even with recent developments in treatment, improvements in the early diagnosis of CRC are essential since they increase survival rates [11].

CRC is leading contributor to morbidity, difficult to treat, and has significant long-term health impacts. Therefore, it is essential to further the study of CRC by considering several viewpoints, such as protein modifications and their role in the emergence of cancer. The significance of protein mutations in CRC that result in altered functionalities will be briefly covered in this review, along with when and how proteogenomics can be used to provide high resolution information on protein status and function and how it can be optimally used to diagnose or treat CRC.

## 2. Colorectal Cancer and Proteogenomics

### 2.1. Mutations That Lead to CRC

CRC tumorigenesis is associated with (1) microsatellite instability (MIN) (2) chromosomal instability (CIN), and (3) mutations involving a wide range of tumor-suppressor genes, proto-oncogenes, and epigenetic changes [12].

Chromosomal instability refers to the loss of one allele which might be associated with tumor-suppressor genes [12]. The genomic changes associated with the chromosomal instability pathway include activation of proto-oncogenes like K-Ras and loss of p53 and Adenomatous polyposis coli (APC) [13]. Microsatellite instability is a consequence of mutations in DNA mismatch repair genes, and that fail to perform repairs during the process of DNA replication. That often results in frameshift mutations, ultimately showing hallmarks of CRC such as angiogenesis and limitless replicative potential which is also the characteristic of stem cells [14,15]. In addition, there are point mutations of different oncogenes associated with CRC. The genes that are susceptible to mutations are *KRAS* [12,16,17], *TP53* [12,18,19], *APC* [12], *BRAF* [12,20], *SMAD4* [12,21–24], *β-Catenin* [12], and *AXIN* [12,25] (Table 1).

**Table 1.** Molecular mutations that lead to colorectal cancer. The genes listed are susceptible to mutations in colorectal cancer along with the locations of genes in the chromosome, the pattern of mutations, and the outcomes.

| Genes | Locations | Function | Mutation Outcomes |
|---|---|---|---|
| *KRAS* | 12p12 | Proto-oncogenes have intrinsic GTPase activity | Trigger the transduction of differentiative signals, even without any extracellular stimuli [12,16,17] |
| *TP53* | short (p) arm of chromosome 17 | Ensures cell cycle arrest and apoptosis to maintain genomic integrity | Results in the formation of a stable protein that no more can bind the DNA and activates target genes [12,18,19] |
| *APC* | 5q21 | Controls transcription of several cell proliferation genes | Increases transcription of β-Catenin targets including cyclin D, ephrins, caspases, and C-myc [12] |
| *BRAF* | 3p22-p21.3 | Proto-oncogene | Results in being constitutionally active in a RAS independent manner [12,20] |

**Table 1.** *Cont.*

| Genes | Locations | Function | Mutation Outcomes |
|---|---|---|---|
| *SMAD4* | long arm (q) of chromosome 18 at band 21.1 | Regulate transcription of target genes, and act as a tumor-suppressor gene | Unable to regulate gene transcription, disrupt TGF-β signaling [12,21–24] |
| *β-Catenin* | 3p22-p21.3 | Transactivate target genes that inhibit apoptosis or encourage cell proliferation | Wnt-signaling activation [12] |
| *AXIN1 and AXIN2* | 16p13.3 and 17q24.1 | Down-regulate WNT pathway | Unable to regulate targeted pathways [12,25] |

This review will focus on the structural abnormalities in the proteins translated from these mutated genes.

### 2.2. Functional Alterations from Molecular Mutations

Molecular mutations in the genes alter the normal functioning of the resultant proteins. RAS proteins can regulate several pathways such as apoptosis, differentiation, and cell growth [26]. Molecular mutations in the *KRAS* genes deregulate the protein to constitutive nature and are active even when there are no external stimuli [17,27]. *TP53* is a transcription factor having pro-apoptotic activity, enabling cell-cycle arrest [28–31]. Mutations in the *TP53* gene lead to a stable mutant protein that fails to bind to the DNA and triggers a set of target genes [19,32]. *BRAF* gene encodes for a protein belonging to the RAF family, and mutations lead to the constitutive activation of the RAS pathway [33]. Targets of *APC* include proteins such as C-myc, cyclin D, caspase, and ephrins. *APC* controls the transcription of these cell proliferation genes [34]. *APC* can also control microtubules [12]. Mutation in the C-terminal sequence of the APC can lead to the deregulation of APC and initiate colon tumorigenesis [35]. *β-Catenin* also transactivates a set of target genes which may induce proliferation of the cells or is inhibitory towards apoptosis [36]. Mutations in the *β-Catenin* lead to its stabilization and ultimately lead to activation of the WNT-signaling [37]. *SMAD* and *AXIN* are also tumor suppressor genes, and mutations of these genes would lead to the activation of CRC [12]. It is also worth mentioning another layer of complexity that complicates CRC is the post-translational modifications (PTM) in the form of glycosylation [38], phosphorylation, acetylation, and ubiquitination [39]. PTMs resulting in amino acid modifications lead to the structural and functional diversity of proteins [40]. In CRC, PTMs can regulate a wide range of cellular processes such as transduction of cell signaling, energy generation and consumption, and DNA reparation [41].

To sum up, we deduce the impact of molecular mutations on the functions of the different players regulating CRC. A current knowledge gap exists on how these mutations can cause structural aberrations in the proteins, which in turn give rise to functional abnormalities. The tumor-suppressor proteins have a target set of genes and understanding the anomalies of the structure-function relationship can be challenging.

### 2.3. Protein Modifications in CRC

2.3.1. Post-Translational Modifications in CRC

Post-translational modification (PTM) has a significant role in the development of cancer [42]. Such alterations can lead to the structural and functional diversity of proteins [40]. In CRC, PTMs like phosphorylation, ubiquitination, and acetylation are modifications of high biological significance [39]. For CRC development, the modifications were identified on the surface of proteins like Plasma protease C1 inhibitor (IC1), vitamin D-binding protein (VDBP), albumin (ALBU), X-ray repair cross-complementing protein 6 (XRCC6), and complement C4-A (CO4A) [39].

2.3.2. Alterations in the Patterns of Glycoproteins and Proteases That Impact CRC Development

Protein glycosylation has received considerable interest in cancer research owing to its relation to cancer development [43]. Plasma glycoproteins have been used to screen different types of cancer such as cancer antigen 125 (CA-125) in ovarian cancer, cancer antigen 15-3 (CA15-3) in breast cancer, and prostate-specific antigen (PSA) in prostate cancer [44–46]. Although, there was a knowledge gap regarding the impact of altered patterns of glycoproteins in CRC until recently [47]. Similarly, cysteine proteases like cathepsin L (CATL) and Cathepsin B (CATB), and serine proteases like tissue-type plasminogen activator (TPA) and urokinase-(UPA) and their inhibitor type-1 (PAI-1) have prominent functions in the CRC development [48]. Such proteins have been upregulated in CRC [48].

The impact of protein glycosylation and proteases is well-known in CRC development, however, there are outstanding questions about the structural alterations in the proteins that might lead to changes in protein functions.

*2.4. Proteogenomics Approaches in Cancer, Specifically in CRC*

CRC is intricately complex and relying only on the information from genomics can be insufficient for cancer diagnosis and treatment. This gives a half-cooked story of the events happening. Information about proteins is welcoming since it can help us to comprehend completely the underlying molecular pathology of cancer [49]. Transcriptomic profiling could be an upgrade that could add to the genome information. Moreover, transcriptomic information could improve the characterization of tumors that could facilitate specific cancer treatments [50]. However, as the target of most anticancer drugs are proteins, the limitation of the transcriptome lies in the fact that it fails to identify the changes in the functional status of the proteins involved in cancer [49]. That said, genomic information could be very useful in deciphering the somatic genomic and epigenomic modifications in the tumor cells [51]. However, there should be cohesiveness in preparing a catalog of these modifications along with systematic functional investigations to uncover the role of these modifications in inducing malignant transformation [51]. One of the large collaborative projects that exist currently is The Clinical Proteomic Tumor Analysis Consortium (CPTAC) [49]. The network initiated by the National Cancer Institute enhanced the comprehension of the molecular basis of cancer [52]. Although CRC is one of the focus areas of this network, it is now populated by ovarian, breast, and other types of cancer as well [52].

PTMs such as ubiquitinylation, phosphorylation, and glycosylation can impact protein stability in CRC [53]. Additionally, PTMs have the potential to alter antibody recognition and affinity. Proteogenomics could yield holistic means to address issues related to CRC by correcting both gene and protein sequences and circumvents the limitations of only genomic and transcriptomic investigations. Proteogenomics approaches in CRC could be used as a useful technique in biomarker discovery. In CRC, searching for predictive biomarkers has been a difficult task [54]. Proper biomarker discovery for a cohort affected by CRC would be helpful for the physicians to recommend specific targeted treatment for that group, thereby reducing the overall health expenses [55]. It has been shown previously that in cancer, the mRNA transcript abundance does not correlate with protein abundance [9]. This study along with several other studies [55–58] highlights the need for proteogenomics in resolving issues associated with cancer. Moreover, these studies also indicate that proteins in CRC are involved in the apoptotic process regulation and cellular protein metabolic process.

The molecular function of proteins is governed by the interaction selectivity with the partner molecules [39]. Such type of interactions often needs a stable and rigid structure of a protein. PTMs may induce small structural changes that would lead to complete loss or switching in the biological activity of the protein. Therefore, the comprehension of the tertiary environment of protein modification is required to understand the function and contributions of such altered proteins in pathophysiological processes.

Identification of global proteomic differences between normal and CRC tumor tissues is necessary to unravel cancer biomarkers and neoantigens. Yet, the current understanding is lacking in CRC cohorts. Global phosphoproteomics, ubiquitomics, and protein glycosylation analyses on human CRC are inadequate at this moment. Phosphoproteomics data suggested that Rb is amplified in CRC and Rb phosphorylation could be a target in CRC [10]. Likewise, phosphoproteomics data from CRC can help in targeting signaling proteins and pathways, providing information to generate therapeutic targets. In CRC, altered ubiquitination of CDK1 could be a pro-metastatic factor in colon adenocarcinoma [59]. Likewise, aberrant protein glycosylation, which resulted in pathological alterations that are widespread in CRC, and the underlying mechanisms for the contributions to CRC tumor progression are largely unknown. Enrichment of genes such as *B3GNT2*, *B4GALT*2, *ST6GALNAC2* has been associated with biosynthesis of N- and cores 1-3 O-linked glycans in the colon, and accounts for 16% of CRCs evaluated [60]. However, such studies are few and need more concerted efforts to understand CRC. Moreover, it is worth noting that the colon cancer-associated proteins and phosphites had little similarities and overlaps with already known cancer genes available in the Cancer Gene Census, further warranting future proteogenomics investigations of proteins, PTMs and associated pathways involved in CRC, so that novel biomarkers and neoantigens are deciphered [10].

*2.5. Information from Proteogenomic Approaches and Precision Oncology*

Proteogenomics is the area of research at the interface of genomics and proteomics [61] (Figure 1).

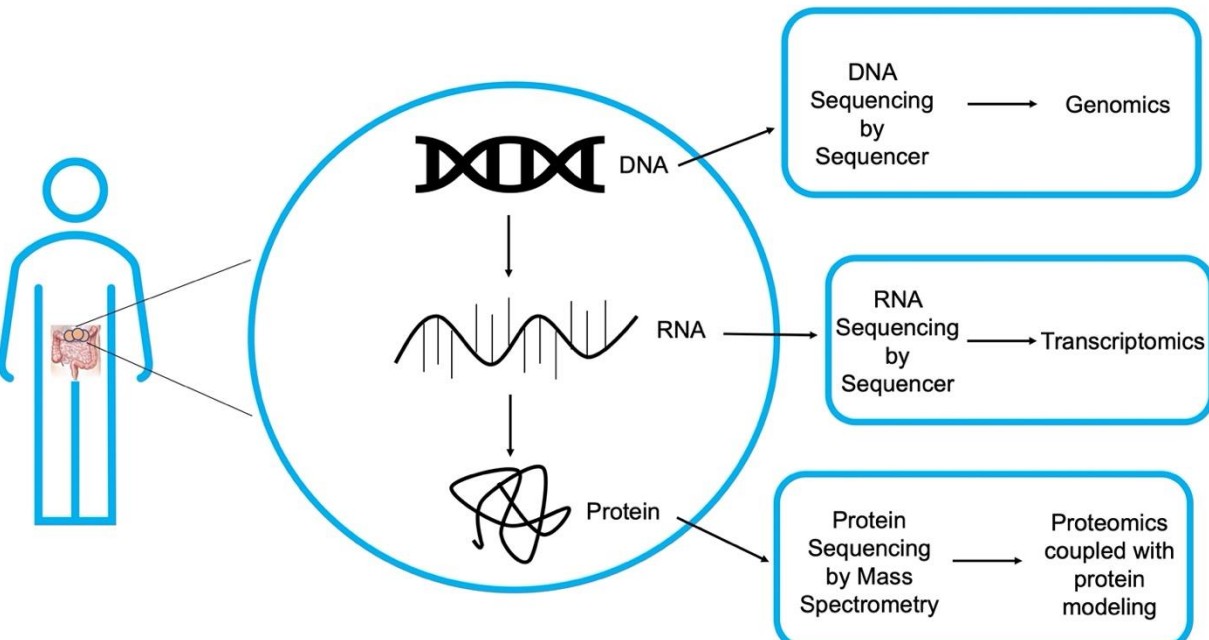

**Figure 1.** The concept of proteogenomics workflow. DNA, RNA and protein extracted from the colorectal cancer patients to be used to understand the genomics, transcriptomics, and proteomics. Protein modeling will provide solution to resolve the structure-function relationship of the proteins involved in CRC.

The field integrates information from genomics, transcriptomics, and proteomics [62]. Proteomics aims at producing a quantitative and complete map of the proteomes for a species [61]. Proteomics can yield a range of information on the cellular localization of proteins, entail details about the protein's interaction partners and networks, post-translational modifications in proteins, whereas phosphoproteomics is able to generate a substantial amount of information about the signaling pathways [63]. This makes proteomics a perfect tool to understand cancer biology as most of the abnormalities will be at the protein levels.

As previously discussed, genomics would not be able to comprehend the events associated with alterations of the protein structures, functions and post-translational modifications. On the contrary, the proteome has the potential to bridge the analytical distance between the cancer genotype and the phenotype [49]. A typical approach to applying the concept of the proteome to cancer is to use the mass-spectrometry-dependent proteomics data and then pairing up with a proper database search algorithm such as MASCOT (Matrix Science Inc., Boston, MA, USA) [49,64]. It is worth mentioning that MASCOT algorithm has been successfully applied to identify novel diagnostic protein biomarkers in CRC [65,66]. There is also the need to include genomics and transcriptomics with proteomics while studying CRC. The introduction of the genomic and transcriptomic information would suffice the limitation of the proteome information alone on the novel proteins that would have no reference in the reference protein database [49]. Genomics, transcriptomics, and proteomics together could capture activity patterns in proteins driven by events like chromosomal deletion and amplification events, DNA copy number, and alterations in the expression of microRNA [49].

This brings us to the discussion of precision oncology—a newer approach to tackling cancer [67]. Precision oncology is the concept of implementing targeted treatments that could be customized based on individual tumor signatures and takes genomic, transcriptomic, and proteomic information into account [67]. There has been significant progress on targeted customized therapies in breast, lung, and melanoma tumors [68]. However, there have been only a few attempts to characterize the CRC using the proteogenomics approach, which could encourage precision oncology [9,10,69]. Till now, proteogenomics of CRC revealed distinct mutational profiles, therapeutic approaches, and candidate driver genes [7,9,10]. In CRC, proteogenomics approaches have the potential to promote customized drug development, however, need improving functional annotation resulting from genomic aberrations [49]. The multidimensional approach is promising for application to precision oncology [9,10,70–74].

Proteogenomics can drive therapeutic hypothesis generation and encourage precision oncology [75]. We propose that predictive biomarkers discovery through proteomics could generate the most efficient drugs to upgrade the current treatment paradigm of CRC. Additionally, newer and novel targets of CRC could provide a rationale for future drug development [75]. However, these approaches in CRC experience a "large p, small n" issue because the number of genes, mutations, proteins, and modification sites that could be evaluated in a study is several times more than the number of samples that can be evaluated in a possible timeline using techniques like LC-MS/MS. These putative candidates for therapeutic applications must be further examined before implementing clinical applications. This review hypothesizes the need for proper, efficient downstream analysis through incorporating proteogenomics approaches coupled with protein modeling.

*2.6. Proteomics to Understand Structures of Proteins*

Combination of Two-dimensional difference gel electrophoresis (2D DIGE) and Matrix-Assisted Laser Desorption/Ionization—Time of Flight (MALDI-TOF) mass spectrometry was successfully applied to identify biomarkers of CRC [76]. Proteins in extracellular vesicles (EVs) of CRC were evaluated using combination of absolute quantification labeling (iTRAQ) [77] and multidimensional liquid-chromatography-tandem mass spectrometry (MS/MS) [78]. Moreover, CRC biomarkers have been successfully identified using liquid chromatography-tandem mass spectrometry (LC-MS/MS) [79]. So, it is evident that mass spectrometry has a successful and considerable application in the identification of proteins associated with CRC.

Native Mass Spectrometry (NMS) has recently evolved and has the potential to be applied in cancer research [80]. NMS can successfully dissect the native contacts between the proteins, protein complexes [81], and protein kinetics [80]. NMS can yield information about protein conformation, cofactor content, topography, and protein complex stoichiometry [82,83]. As reported before, CRC is due to mutations of oncogenic mutants including

*KRAS*. *KRAS* mutations are prevalent in CRC amounting to 45% in the United States and as high as 49% in China [84]. NMS could be ideal to study energetics and kinetics of GTP hydrolysis of such oncogenic mutants and makes NMS suitable particularly for CRC [80]. KRAS protein is a membrane-bound GTPase (GTP hydrolase) which can control a range of cellular signaling pathways like the MAPK and PI3K pathways [84]. When bound to the GDP form, KRAS remains in an inactive state, and upon GTP-bound, the protein becomes active. In CRC oncogenic status, it is often observed that the KRAS exists in active status as its intrinsic GTPase function and GTPase activating proteins are hindered [85,86]. So, GTP hydrolysis status is key in CRC patients, and the effectiveness of NMS could help in the detailed investigation of such GTP hydrolysis. Mutant alleles with strongly hindered GTP hydrolysis including G12R and Q61R/K/L mutants, are less effectively degraded by direct pan-KRAS proteolysis targeting chimeras (PROTACs) targeting the GDP-bound form [84]. More investigations into such and other mutation forms would potentially help to establish newer KRAS-targeting drugs as a concept of precision therapy and can substitute for existing chemotherapeutic approaches.

Hydrogen/deuterium exchange (HDX) is another technique that can reveal substantial information about the dynamics of the protein [81]. HDX-MS thus has the potential to be applied in elucidating the PTMs associated with CRC by giving a clear account of the conformational dynamics of proteins and protein complexes [87]. Although this technique is nascent in being applied to the field of cancer investigation but could be beneficial if it could be applied. HDX-MS can easily bridge the existing knowledge gap of the protein structure-function relationship [81].

We emphasize on proteogenomics approach that uses the information from DNA sequencing, expressed sequence tags (ESTs), RNA sequencing, and ribosome profiling to generate customized protein sequence databases. This information would then help to interpret the proteomics data from MALDI-TOF-MS, LC-MS/MS, and iTRAQ-LC. This customized data would then encourage validation at the protein level of the gene expression data, and in turn refine gene models [61].

A typical proteogenomics approach workflow involves the extraction of DNA, RNA, and proteins from the tumor samples [88]. The DNA extracted will be used to obtain the genomic information, mRNA for transcriptome profiling, and proteins to understand the proteome [89]. For DNA and RNA sequence data generated, the first approach will be to trim the low-quality sequences. DNA data could be used to detect mutations, somatic copy number alterations, and microsatellite instability prediction, while RNA data can be used to obtain transcriptome profiles [89]. Workflow for proteome involves extraction of proteins, tryptic digestion, labeling of proteins by TMT-10, peptide fractionation by liquid chromatography, metal affinity chromatography to perform the phosphopeptide enrichment, and implementation of a mass spectrometry technique such as LC-MS/MS [90]. High-Performance Liquid Chromatography and a combination of Mass Spectrometry (HPLC-MS/MS) have been successfully applied to identify and characterize the PTMs associated with CRC when compared with normal patients [39]. Specifically, a bioinformatics approach was put forward for CRC patients and compared with healthy samples to identify acetylation, phosphorylation, and ubiquitination which eventually aimed at determining the altered biological activity of the proteins [39]. Acetylation, phosphorylation, and ubiquitination have significant contributions to CRC. Phosphorylated retinoblastoma proteins (Rb) have been linked with reduced apoptosis in CRC cells [91]. Phosphorylated Rbs interact with different protein partners and show different Rb functions in CRC [91]. Protein lysine acetylation could impact CRC metastasis through several pathways. PTMs involving proteins like isocitrate dehydrogenase (IDH1) are seen to influence hypoxia-inducible factor 1-alpha (HIF1$\alpha$) dependent transcription of steroid receptor co-activator (SRC) transcription which further control CRC progression [92]. A very recent approach for cancer treatment is to eliminate oncogenic proteins by regulating the ubiquitin-proteasome system (UPS) [93]. Deubiquitinating enzymes (DUBs) tend to play important role in CRC formation and development by increasing the oncogenes stability [94]. Specifically, in CRC,

the Wnt-signaling pathway has been observed to be impacted by USP14 by enhancing Wnt signaling pathway [95]. USP4 and USP7 enhance the β-catenin and results in CRC tumor progression by modulating Wnt signaling [96,97]. This reiterates the study of such PTMs in CRC and forms the theoretical background for the application of such protein PTMs.

### 2.7. Approach to Protein Structure Modeling, and In Silico Mutations

One of the resources the scientific community has is the rich depository of crystal structures of the various tumor-suppressor proteins [98–101]. This review proposes to optimally utilize the crystal structure depositories so that the investigators can generate the necessary tertiary structure of various proteins linked to CRC. One popular way to generate the wild-type and mutated structure of the proteins is to use the power of protein structure modeling and in silico mutations using the information from these protein crystal structure depositories such as RCSB Protein Data Bank.

Protein mutations can lead to six possible outcomes—(1) protein activity alterations [102–107] (2) impacting protein–protein interactions [108–111] (3) affecting protein folding [103–105] (4) modifications in protein localization [112,113], (5) changing the half-life [114], and (6) combination of all these effects [102,103,108,109,112–114]. The knowledge from protein structure modeling and in silico mutations can address the knowledge gaps involving altered protein activity, structural abnormalities, protein–protein interactions, and protein folding.

Protein modeling, molecular dynamics simulation, and in silico mutations have been successfully employed to gain insights into the tumor-suppressor proteins in various cancer studies [115–117]. Additionally, this approach has even been applied to proteins involved in CRC [118–120]. However, there should be more concerted efforts to apply these in silico approaches to solve the mysteries involving cancer, and with a pool of proteomics data coming up, that should provide an ideal platform to implement this concept in near future [121].

### 2.8. Protein Modeling to Assess the Tertiary Structure (3D) of Proteins in CRC

Mass Spectrometry over the years has yielded a wealth of information on the identification of biomarkers of CRC [122,123]. Biomarkers can differentiate between a healthy patient with a cancer patient as biomarkers are only detected in the patient's blood or body fluids [124]. However, we want to highlight the importance of the structural aberrations in the proteins associated with CRC. The proteogenomics approach will generate insights into the gene levels, mRNA levels, and protein levels as well. A well-planned approach to applying mass spectrometry and in silico protein modeling could open new frontiers in cancer research, including CRC. Protein modeling would provide insights into the mechanisms of the functions of the altered proteins [125].

The usual approach from the mass spectrometry technique is to generate a spectrum from which peptide sequences can be detected [126]. Then, the sequences are searched against the NCBI database to identify the candidate proteins from which the sequences have been obtained [126]. As discussed previously, this approach could be problematic when novel proteins need to be identified. The absence of the necessary information in the database might pose challenges to identify those novel candidates. This is where proteogenomics could be useful, and conjugating proteogenomics with protein modeling would not only identifying the novel proteins but could also entail insights into the detailed mechanisms of such proteins [49,127].

Future cancer research investigations should focus on the tertiary structures of the proteins rather than on the sequences. The peptide sequences coming out of the mass spectrometry studies could be used as an input for the protein modeling techniques. One of the typical workflows could include the involvement of standard crosslinking techniques in which a crosslinker reacts with a definite residue and the second one form the crosslink [128]. A crosslinker is a reagent that has two functional groups separated by a spacer region. Two types of cross-linkers could be used for protein structural modeling—(1)

Standard crosslinkers, (2) Photo-crosslinkers. The crosslinker form covalent bonds when the crosslinker reacts with protein. For using photo-crosslinkers, the photoreactive groups are required to be activated with ultraviolet light [128]. The user then needs to comprehend the upper distance bound of the involved crosslinked residues. The two residues of the proteins can only react if the distance of the reactive groups is within the range of the crosslinker. The reactive crosslinkers would then store the spatial information. To elucidate the spatial information, the experimenter then digests the protein using enzymes like trypsin or other proteases [128]. Then, mass spectrometry of the generated peptides is carried out. After that, specialized and customized database search software evaluates the crosslinks resulting from the mass spectrometry data. The crosslinks finally dictate the input data to data-driven protein structure modeling [128].

Primarily, three protein modeling methods exist currently (Table 2):

(1) Homology modeling: This method also known as comparative modeling could be used when a protein with a crystal structure is available in the database [129]. The query protein must possess >30% sequence identity with the protein available in the database [130]. The homology model could be built using efficient tools like MODELLER [131]. Previously, a wide range of proteins associated with cancer has been studied by this method [132–134].

(2) Modeling by threading/fold recognition: Information on the protein folds based on similar proteins is used in predicting the structure of the proteins the users want to model. I-TASSER online server [135–137] can be used for modeling where different databases are used and the workflow is user-friendly.

(3) Ab initio strategy: This is a powerful approach to predict protein structures when an appropriate homolog structure is unavailable in the database. The model is initiated and built using the information on the most favorable energy conformations of the participating amino acids, and also calculates the potential chemical interactions among the amino acid sequences [138]. However, this technique can be time-consuming and computationally intensive [139]. I-TASSER can apply the ab initio modeling when an appropriate template is absent [135]. QUARK [140,141] and CONFOLD2 [142] are other useful web servers for generating ab initio protein structures from amino acid sequences. While QUARK uses Monte Carlo simulation under the influence of an atomic-level-knowledge-based force field, CONFOLD2 uses a subset of input contacts to understand the protein fold space guided by a soft square energy function.

**Table 2.** Three protein modeling strategies with advantages and disadvantages of each system.

| Protein Modeling System | Advantages | Disadvantages | Reference |
|---|---|---|---|
| Homology modeling | High-resolution structures can be generated | Physicochemical principle of protein modeling cannot be deciphered | [143] |
| Modeling by threading/ fold recognition | Works better for proteins when templates available are of distant homologies | The structures are less reliable than homology modeling | [144] |
| Ab initio strategy | Answers on how the protein takes a specific structure out of many structural possibilities | Less reliable for larger protein structures composed of more than 150 residues | [145] |

To model the proteins with PTMs, PyTMs, a plugin devised for PyMOL is very useful [146]. The workflow enables easy standardization techniques, and ensures fast and easy user controls [146].

Although the techniques are independent, these can be intertwined to solve the protein structures. For example, the homology modeling approach and ab initio strategies could be used simultaneously. The ab initio can predict the areas of the proteins without homology, while the homology modeling approach can be applied to the other parts of the protein where there is a template to work with [81]. The generated structure can then be validated

using chemical cross-linking coupled to mass spectrometry (XL-MS) and surface labeling coupled to MS (SL-MS) [81]. The downsides of the protein modeling approach are its incapability of predicting multimeric protein complexes [81]. The downside however can be solved by using protein docking tools that can predict the models of protein complexes [81].

Often studies related to cancer might demand insights into various protein complexes, and so a definite strategy should be important. In CRC development, it is often observed that multiple candidates or protein complexes are involved either independently or in cascades, and so a definite strategy should be in place to deal with such instances [147]. Even the involvement of protein complexes in chemotherapy resistance for CRC has been observed [148]. However, such findings on the involvement of multimeric protein complexes and cascades are still in infancy and the involvement of proteogenomics approaches could be useful to understand CRC more extensively. A tool like ClusPro2.0 utilizes a thermodynamics-based approach to predict the lowest energy well for interacting proteins [149]. During the protein–protein docking, one protein is kept static and another forms several conformations. Similar conformations are kept together, and the conformations with the highest frequency are selected. The selected candidates are then refined and energy-minimized, and could be further validated using solution techniques like SL-MS and XL-MS [149]. If multiple subunits are required to generate a protein complex, this strategy can be repeated several times to consider additional subunits one at a time.

In cancer, cofactors often play roles in protein functioning and these need to be considered while generating protein 3D structures [150,151]. A usual approach could be to use SwissDock to locate the cofactor position within the protein and then perform the energy minimization to obtain the best possible confirmation of the protein with the cofactor [152].

For visualizing the monomeric and multimeric proteins, Chimera is a productive software with the plugin Xlink Analyzer that analyzes the protein structure validation using cross-linking data [153,154].

The target sites of colon cancer-related proteins are often nucleic acids, which then guide the downstream processes [155]. Modeling such protein-nucleic acids can be challenging, although there have been some recent breakthroughs [156,157]. Structure-based methods are preferred over sequence-based methods, and PRIME 2.0.1 is useful in predicting such complex protein-DNA structures [158].

*2.9. Protein Docking of Mutated Proteins to the Substrates to Understand the Impact on Functions, the Need for Energy Minimization and Molecular Dynamics Simulation*

Proteins that have structural alterations should have a profound effect leading to cancer [159]. So, understanding the mutational implications of the structural stabilities of the proteins is critical. There are reports of understanding the mutants of the key proteins [160,161], however, there should be more initiatives from the scientific community that could decipher the mutated protein's functional mechanisms. One approach to induce amino acid alterations is to use Swiss PDB viewer [162], and then use appropriate protein docking software to dock the involved partners of the mutated proteins. There are several protein docking tools available currently with Autodock [163] and Autodock Vina [164,165] being the most reliable ones. The users will be able to easily compare the wild-type protein with the mutated counterparts and how the mutation impacts the overall binding affinities or specificities with the interacting partners. The change in the functions could also be tracked by such an approach [161].

Energy minimization, structural refinements, and molecular dynamics simulations are critical steps that need to be followed during protein molecular docking. Energy minimization of protein structures ensures the proper molecular arrangement of amino acids in space as the initial structures might not be energetically favorable [166]. Structural refinements of the protein structure are then carried out by adding missing atoms and neutralizing charges [167]. Energy minimization and structural refinements are steps under molecular dynamics simulation studies. Trajectory data from simulation should be

analyzed carefully to select the best energy-minimized structure of proteins, interacting partners and the protein-ligand structures [167]. Energy-minimized refined structures with appropriate simulation techniques ensure the stability and near-naiveness of the protein structures and interactions [168]. Currently, several useful tools are available to perform the simulation of proteins, with AMBER [169], NAMD [170], GROMACS [171], and CHARMM [172] being the most popular ones.

*2.10. Combining Hydrogen Exchange Mass Spectrometry and Protein Modeling to Understand 3D Structures*

Hydrogen deuterium exchange mass spectrometry (HDX-MS) is a sophisticated and powerful technique that can elucidate the behavior of proteins by resolving the structure, dynamics, and function [173]. One of the interesting ways to optimize the HDX-MS experimental data is by conjugating the data to guide computational modeling tools like molecular docking and molecular dynamics [174,175]. In the past, molecular dynamics simulations have been used to decipher the structural properties obtained from HDX-MS data [176–182], even complementing the HDX-MS data by applying short time scales [183–186]. So going forward, there is a huge scope for integrating the strength of HDX-MS and modeling techniques to study the protein structure alterations in CRC [187].

**3. Conclusions**

Studies carried out in microorganisms like bacteria and yeast indicated a 50% correlation at the mRNA and protein levels [188]. However, the correlation significantly decreases when the genome complexity increases as in the case of humans, where only 30% of variations in protein levels could be explained by corresponding changes in the levels of mRNA [189]. This is due to various post-transcriptional (variable mRNA translational efficiency, siRNA regulation) and post-translational regulations (phosphorylation, glycosylation) which impact the protein stability [190]. So, it is difficult to comprehend the proteome dynamics from functional genomics data alone, and there is a need to complement the proteomics approach to genomics and transcriptomics resulting in the complete proteogenomics approach [190].

CRC is one of the deadliest cancers in the world [191]. Human genome sequencing has enabled the scientific community to fathom the genetic alterations in CRC in a comprehensive manner [192]. However, interpreting CRC only at the genomic level will not give the full picture, instead, the review suggests incorporating the information from the RNA and protein levels to gauge the functional alterations.

Proteogenomics approaches in CRC can facilitate the concept of precision oncology and encourage the applications of customized targeted treatments. In CRC, the proteogenomics approaches have been successfully applied to reveal new therapeutic avenues [10]. Specifically, personalized neoantigens for CRC patients were identified that could be used for the generation of therapeutic hypothesis [10].

The review highlights the importance of proteomics, and the way it could be integrated with protein modeling to grasp the underlying knowledge gaps in understanding the structural alterations in CRC, and also cancer in general. The major takeaways are: (1) From the clinical perspective, CRC is caused due to genetic changes involving *KRAS*, *TP53*, *APC*, *BRAF*, *SMAD4* and several other alterations (2) These alterations are not only a phenomenon occurring at the genomic or transcriptomic level, but does show up at the protein level owing to mutated structures and changed functions of such proteins (3) So, it is the best bet to correlate the genotype-phenotype attributes to capture the full extent of the CRC diagnosis, biomarker discovery and treatment (4) Proteomics has the potential to resolve the existing issues of understanding CRC by providing new insights regarding protein profiling, biomarker discovery and PTMs (5) The MS-based proteomics data would be coupled with protein modeling, protein docking, in silico mutations to understand the differences in protein's behavior between healthy and CRC patients. This knowledge would

then complement the information from clinical studies that relies on targeted proteomics to yield a better understanding of CRC. We not only discussed the existing information and questions associated with understanding CRC but also provided the avenues that can be explored to meet the current challenges.

**Funding:** This research received no external funding.

**Institutional Review Board Statement:** Not applicable.

**Informed Consent Statement:** Not applicable.

**Acknowledgments:** We are grateful to Sonny TM Lee and Tanner Richie of Division of Biology, Kansas State University for critically reviewing and editing the article. We would like to acknowledge the Lee lab, Division of Biology, Kansas State University for helpful insights, inputs, and discussions. Soumyadev Sarkar would also like to acknowledge the support of Ferran Garcia-Pichel of Center for Fundamental and Applied Microbiomics, Arizona State University and all the members of Garcia-Pichel Lab. The authors are thankful to Srijoni Basu and Bidisha Ghosh for the help with the inputs for the preparation of the figure.

**Conflicts of Interest:** The authors declare no conflict of interest.

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
