# Peer review of "Proteogenomic Approaches to Understand Gene Mutations and Protein Structural Alterations in Colon Cancer"

_physiologia, doi:10.3390/physiologia3010002_

Round 1

Reviewer 1 Report

GOOD EFFORT.I FIND POOR THE SECTOR OF CONCLUSIONS.OR SHOULD BE MORE EXPLAINATORY EITHER TO WRITE A DISCUSSION WHERE TO EXPLAIN THE REASON WHY THE PROTEOGENOMICS ARE "BETTER"THAN THE "SINGLES" OMICS.FURTHERMORE A REFERANCE TO THE STAGE OR THERAPY IT WOULD BE ADDED.

Reviewer 2 Report

The manuscript authored by Soumyadev Sarkar aims to address the protegenomics-based approaches in CRC. The review starts by listing genetic mutations associated with CRC and then mentions the affected proteins. Moreover, it tries to describe also the CRC-associated PTMs, with a dedicated paragraph listing some cancer antigens, although the PTMs-relevance in the proteogenomic context is not described. 

Further, there is a dedicated paragraph to proteogenomics approaches in CRC, but this contains only some general notions about genomics and transcriptomics but not the main particularities of proteogenomics in CRC. More specifically what is the difference in using proteogenomics in CRC, compared with other cancers? In my opinion this should have been the focus of this chapter and such a description would result in a greater interest to the reader, particularly considering the clinical context of CRC. 

Next, most of the manuscript text lists various algorithms used either in mass spectrometry data analysis or protein structural investigation, but their relevance in the context of CRC proteogenomic is not described. The manuscript looks more like a description of CRC mutations and various algorithms rather than a hypothesis driven review on how proteogenomics could advance the biomedical field in CRC. Thus I cannot recommend publication. Below are some points to consider:

Major points:

1. The author states that “…significance of protein mutations in CRC that result in altered functionalities will be briefly covered in this review, along with when and how proteogenomics can be used to provide high resolution information on protein status and function and how it can be optimally used to diagnose or treat CRC.”. However the review fails to address exactly the problem that it tries to resolve. First, the author only lists mutations linked to CRC, but does not describe the exact strategy on how proteogenomics could represent a solution to this problem. What would be the major key-points in a particular clinical context that proteogenomics would address?

2. P5, L178-180: The author states: “A typical approach to applying the concept of the proteome to cancer is to use the mass-spectrometry-dependent proteomics data and then pair it with the mapped peptides that already existed in the reference protein data-base”. This sentence has no logic and should be modified. Is the author referring to the database search algorithm or to the possibility of combining the mass spectrometry results to other MS-published data in repositories?

3. P6, L214-216: How would NMS and HDX be used in CRC precision oncology? What makes these suitable, particular for CRC?

4. P6, L235-236: Is this referring to PTMs which are found in CRC compared to normal?

5. P7, L279: How could the linear peptides identified in a typical shotgun proteomics be used for protein modeling? This is not described.

6. P8, L313-326: How does all of these are related to CRC proteogenomics?

Minor points:

1. P1, L40-41: Repetitions should be avoided “… are indirectly caused by a number of causes …”.

2. P5, L172: Signaling pathways are usually characterized using phosphoproteomics.

3. P6, L203-205: There is no iTRAQ-LC. iTRAQ is a peptide labeling method used for multiplexing in quantitative proteomics. Thus, the author should state if it referrers to the label-free quan or the labeled one.

4. P6, L205: Tandem mass spectrometry is MS/MS not MS!.

5. The main three protein modeling systems should be described in a table with pros and cons to be much easier for the reader to assess the differences between these.

Round 2

Reviewer 2 Report

The author submitted a revised version of the manuscript addressing only modestly my initial concerns. I find the review to be too vague, using only some general considerations, without addressing properly what should supposed to address: proteogenomic in CRC pathology. The author just lists lots of concepts, techniques used in different contexts without addressing their logical motivation in CRC. Thus, I cannot recommend publication. I have described below some points which I find most important:

1. My initial assessment was related to the PTM relevance in the context of proteogenomics in cancer. The author indeed added a paragraph listing some of the most well-studied PTMs (phosphorylation, glycosylation, ubiquitination) but does not describe what is the relevance of this information in the context of proteogenomics in CRC and how this extra data could be used practically in such experiments. The author cites: "Often, the half-life of proteins is governed by ubiquitin and proteome pathways". There is a lot of regulation regarding protein degradation and for sure the canonical ubiquitin proteasome pathway is not the only one to consider as there are alternative routes of protein degradation such as lysosomal, autophagosomal etc. The paragraph looks like it mixes a lot of things siRNA regulation, protein degradation, PTMs etc. Thus, in my opinion it confuses the reader rather than to elaborate the subject it tries to address. 

2. The added paragraph which supposedly address the particularities of proteogenomic applications in CRC refers to considerations which are generally valid in cancer pathology and not necessarily particularly only to CRC. Also, the authors mixes some concepts from fundamental studies to concepts valid only in a clinical context. For example, what would be the reasoning of performing “an in-depth investigation of pathways involved in CRC development specifically at the protein level”? Does the author refers to biomarker discovery or to the use of biomarkers in a clinical context for patient-specific treatment monitoring? Also, why in the author’s opinion the low correlation between mRNA and protein abundance would be a reason for to use proteogenomic in CRC? This limited correlation has been observed previously for other biological systems, so it is not necessarily particular for CRC.

3. The authors states that introduced the hypothesis which could advance the biomedical field in CRC. Unfortunately, the hypothesis is missing and frankly it is sad to see inserted an almost similar phrase with the title of the article given as a reference. In a review usually the authors introduce new concepts and ideas which could move forward the knowledge in a particular field and not just lists some algorithms without addressing their proper relevance to the CRC pathological context.

4. The authors states: “Proteomics has the potential to resolve the existing issues of understanding CRC by providing new in-sights regarding protein profiling, biomarker discovery, and PTMs. The MS-based proteomics data would be coupled with protein modeling, protein docking, in silico mutations to understand the full extent of normal and CRC induced altered protein’s behavior.” Does the authors expects to see “protein modeling, protein docking, in silico mutations to understand the full extent of normal and CRC induced altered protein’s behavior” in every clinical facility? It is surprising for me to see such a strategy as usually these are concepts and techniques used mostly in hypothesis driven studies rather than in laboratories from the clinics side which focuses more towards targeted proteomics and not discovery based one. 

5. What does the author define as “a proper database search algorithm”? There are various algorithms currently available starting from the oldest ones like Sequest and Mascot and going to the more recently DIA-based used algorithms like Spectronaut or DIA-NN. The phrase added introduces more confusion to the reader actually, as it is quite vague.

6. Why would GTP hydrolysis of KRAS oncogenic mutants be relevant to CRC precision oncology? How would help the patient, practically speaking? It is quite confusing to understand the utility of such approaches only from this phrase.

7. Why only acetylation, phosphorylation and ubiquitination are the only relevant PTMs in CRC patients? What was the conclusion of such studies? Just mentioning them does not bring any value to the manuscript in my opinion.

8. It is interesting to note that I have asked about the linear peptides but the author refers to cross-linked peptides in the provided answer. However, there are additionally considerations even for cross-links experiments. Which crosslinkers? Are all crosslinkers fitted for protein structure modeling? It is vague and general the paragraph inserted by the author.
